# Optimization and Standardization of Human Saliva Collection for MALDI-TOF MS

**DOI:** 10.3390/diagnostics11081304

**Published:** 2021-07-21

**Authors:** Monique Melo Costa, Nicolas Benoit, Florian Saby, Bruno Pradines, Samuel Granjeaud, Lionel Almeras

**Affiliations:** 1Unité Parasitologie et Entomologie, Département Microbiologie et Maladies Infectieuses, Institut de Recherche Biomédicale des Armées, 13005 Marseille, France; mcosta.monique@gmail.com (M.M.C.); nicobenoit73@hotmail.com (N.B.); florian.saby@etu.univ-amu.fr (F.S.); bruno.pradines@gmail.com (B.P.); 2Aix Marseille University, IRD, SSA, AP-HM, VITROME, 13005 Marseille, France; 3IHU Méditerranée Infection, 13005 Marseille, France; 4Centre National de Référence du Paludisme, 13005 Marseille, France; 5CRCM Integrative Bioinformatics Platform, Centre de Recherche en Cancérologie de Marseille, INSERM, U1068, Institut Paoli-Calmettes, CNRS, UMR7258, Aix-Marseille Université UM 105, 13009 Marseille, France; samuel.granjeaud@inserm.fr

**Keywords:** SARS-CoV-2 diagnosis, saliva, MS profiling, standardization

## Abstract

SARS-CoV-2 outbreak led to unprecedented innovative scientific research to preclude the virus dissemination and limit its impact on life expectancy. Waiting for the collective immunity by vaccination, mass-testing, and isolation of positive cases remain essential. The development of a diagnosis method requiring a simple and non-invasive sampling with a quick and low-cost approach is on demand. We hypothesized that the combination of saliva specimens with MALDI-TOF MS profiling analyses could be the winning duo. Before characterizing MS saliva signatures associated with SARS-CoV-2 infection, optimization and standardization of sample collection, preparation and storage up to MS analyses appeared compulsory. In this view, successive experiments were performed on saliva from healthy healthcare workers. Specimen sampling with a roll cotton of Salivette^®^ devices appeared the most appropriate collection mode. Saliva protein precipitation with organic buffers did not improved MS spectra profiles compared to a direct loading of samples mixed with acetonitrile/formic acid buffer onto MS plate. The assessment of sample storage conditions and duration revealed that saliva should be stored on ice until MS analysis, which should occur on the day of sampling. Kinetic collection of saliva highlighted reproducibility of saliva MS profiles over four successive days and also at two-week intervals. The intra-individual stability of saliva MS profiles should be a key factor in the future investigation for biomarkers associated with SARS-CoV-2 infection. However, the singularity of MS profiles between individuals will require the development of sophisticated bio-statistical analyses such as machine learning approaches. MALDI-TOF MS profiling of saliva could be a promising PCR-free tool for SARS-CoV-2 screening.

## 1. Introduction

Saliva has multiple roles in the maintenance of oral health, including protection against dental demineralization and microbial invasion, tissue lubrication, and food digestion [1]. In addition to oral mucosal proteins, hundreds of plasma proteins have been identified in human saliva [2]. The proteins present in saliva come from both local and systemic sources. The composition of saliva is very informative for analyzing and comparing the physiological or pathological state of individuals [3]. Consequently, all lesions, pathologies, and infections with oral involvement make the detection of salivary biomarkers particularly attractive. Currently, salivary biomarkers help to detect oral cancer, dental caries, periodontal disease, but also other pathologies with non-oral tropism, such as diabetes, breast cancer or even lung cancer [4,5]. Recent proteomic studies have provided convincing results for the use of human saliva for diagnostic and prognostic purposes [6].

Among the strategies used for salivary biomarker screening, Matrix-assisted laser desorption ionization time-of-flight mass spectrometry (MALDI-TOF MS) profiling has consistently shown its ability to classify individuals according to their infectious status, notably for oral pathologies (e.g., dental caries, periodontitis, oral lichen, mouth cancer), based on saliva analysis [7,8]. In addition, other work has demonstrated the potential of MALDI-TOF MS tool for identifying respiratory viruses from cell culture supernatant [9].

MALDI-TOF MS profiling is used for more than a decade in routine diagnostics by microbiology laboratories for the identification of microorganisms (e.g., bacteria, yeasts, and fungi) [10]. This tool has revolutionized identification techniques in microbiology due to its rapidity, simplicity of sample preparation and low cost [11]. The principle of sample classification relies on matching the MS spectra of the query with a library of reference MS spectra. This spectral matching strategy was successfully applied for the identification of viral species [9,12].

The high performances of MALDI-TOF profiling for the identification of viruses and determination of infectious status using saliva make this tool a promising diagnostic approach in the current world health situation. Indeed, the emergence of a novel coronavirus causing the severe acute respiratory syndrome coronavirus 2 (SARS-CoV-2), from Hubai province, China, in December 2019 has required extensive population screening for pandemic management. Although RT-qPCR from nasopharyngeal swab (NPSs) remains the reference strategy for the diagnosis of SARS-CoV-2, the repeated demonstration of detection of this virus in the human saliva has underlined the value of this body fluid for the diagnosis of coronavirus diseases [13,14]. A high concordance rate in the detection of respiratory viruses using molecular assays [15] between saliva and nasopharyngeal samples, including coronaviruses, has been demonstrated [16]. Although the clinical diagnosis of Coronavirus Disease 2019 (COVID-19) using saliva samples remains controversial, the comparison of nasopharyngeal and saliva paired-samples showed that SARS-CoV-2 was detected equivalently or better in saliva [17,18,19]. The identification of SARS-CoV-2 proteins in gargle solutions from patients with COVID-19 infection by a MS-method confirmed the local presence of viral proteins [20]. Moreover, the high expression of the main host receptor of SARS-CoV-2, the angiotensin-converting enzyme 2, on the epithelial cells of tongue and salivary glands, explains the detection of the virus in oral fluids [21]. 

Based on these studies, we hypothesized that MALDI-TOF MS profiling of saliva could be an interesting alternative to molecular assays for the diagnosis of SARS-CoV-2. Before identifying a MS protein signature associated with SARS-CoV-2 infection in the saliva of COVID-19 patients, we evaluated the reproducibility and stability of MS profiles among healthy individuals according to saliva sample collection and management. In this context, standardized and optimized procedures for the collection and preparation of saliva samples for MALDI-TOF MS analysis were established. 

## 2. Materials and Methods

### 2.1. Ethical Statement

The study protocol was reviewed and approved by the Ile de France 1 ethical committee (no. 2020-A01249-30 protocol, 6 August 2020). Demographics data and samples were collected uniquely after the participant understood the study protocol and acknowledged informed consent. All participant information and samples were anonymized before use.

### 2.2. Participant Enrollment

Healthcare workers without anosmia/ageusia, flu symptom or respiratory disorder, for at least two weeks before sampling, were invited to enroll in the study. All participants with specific symptoms of COVID-19 were invited to perform a diagnostic PCR for SARS-CoV-2 from NPSs as previously described [22] and positive individuals were excluded from the study. Individuals under 18 years old, non-French speaking, pregnant women and individuals suffering of Gougerot-Sjögren Syndrome were excluded. A total of 12 individuals were included. The median age was 40.1 years (range, 27–62 years) and eight individuals (66.6%) were men.

### 2.3. Saliva Collection

Saliva was collected in the morning between 8:00 am and 10:00 am. When saliva was collected at other times, this was specified. A bottle of spring water (Cristaline, Cairanne, France) was given to each participant who performed a quick mouthwash to remove drink and food residues before saliva collection. Two saliva collection procedures were tested.

#### 2.3.1. Drooling in Plastic Tubes

Saliva samples were collected in a sterile 50 mL plastic tube by spiting the saliva in order to obtain approximately 1–2 mL of sample. The samples were self-collected by passive drooling and instructions were provided to each participant to avoid mucous secretions from oropharynx or lower respiratory tract (i.e., sputum). Immediately after the saliva collection, 1 volume of ultra-pure water was added to facilitate pipetting of the samples which were then vortexed and centrifuged (2 min at 1500× *g*) to pellet cell debris. Saliva samples were stored on ice until use.

#### 2.3.2. Cotton Roll System

Neutral Salivettes^®^ (Sarstedt, Numbrecht, Germany), with cotton roll free of component to stimulate salivation, were used according to the manufacturer’s instructions. Briefly, the cotton roll was directly introduced in the mouth without handling and kept 2 min in the mouth of the participant who soaked the cotton with circular movements, prior to replace it into the stopper part of the Salivette tube. The samples were stored in ice until their use. All Salivettes were centrifuged at 1500× *g* for 2 min at 4 °C and the recovered saliva was transferred to 1.5 mL Eppendorf tubes and stored at 4 °C until use.

### 2.4. Saliva Sample Preparation

Saliva collection with plastic tubes were used to determine the best sample preparation condition before MS analysis by comparing four protocols. 

Protocol A: Saliva was mix with one volume of ultra-pure water, vortexed and centrifuged 2 min at 1500× *g* to pellet cells and cell debris. One microliter of supernatant was directly loaded in quadruplicate on the MALDI target plate (Bruker Daltonics, Wissembourg, France).

Protocol B: Saliva was centrifuged 2 min at 1500× *g*. Twenty microliters of saliva supernatant were mixed with one volume (i.e., 20 µL) of the Mix buffer (mixture (50/50) of 70% (*v*/*v*) formic acid and 50% (*v*/*v*) acetonitrile (Sigma, Lyon, France)) used for MALDI-TOF MS profiling analysis [23]. One microliter of saliva mix was then loaded on the MALDI target plate in quadruplicate.

Protocol C: Saliva was centrifuged 2 min at 1500× *g*. One hundred microliters of saliva supernatant were mixed with 400 µL of ice-cold absolute ethanol (100%). The mixture was vortexed, incubated overnight (ON) at −20 °C and centrifuged at 15,000× *g* at 4 °C during 30 min. The supernatant was discarded and the pellet was dried at room temperature (RT) during 30 min. The pellet was re-suspended with 100 µL of the Mix buffer (mixture (*v*/*v*) acetonitrile 50%/formic acid 70%) by three successive cycles of vortex and pipette homogenization. One microliter of saliva mix was then loaded on the MALDI target plate in quadruplicate.

Protocol D: The procedure described for protein precipitation by ethanol was repeated except that the ethanol was replaced by ice cold acetone (100%, Sigma, Lyon, France). Briefly, 100 µL of saliva sample were mixed with 400 µL of ice-cold acetone (100%). The mixture was vortexed, incubated ON at −20 °C and centrifuged at 15,000× *g* at 4 °C during 30 min. The supernatant was discarded and the pellet was dried at RT during 30 min. The pellet was re-suspended with 100 µL of the Mix buffer (mixture (*v*/*v*) acetonitrile 50%/formic acid 70%) by three successive cycles of vortex and pipette homogenization. One microliter of saliva mix was then loaded on the MALDI target plate in quadruplicate.

### 2.5. Serial Dilution of Saliva

Fifty microliters of supernatant from saliva collected with plastic tube, as described above, were transferred in a new tube and was serially diluted by half with ultra-pure water until 1/256 were done. The final dilution was performed by adding one volume of Mix buffer to each diluted sample prior loading 1 µL of the mixture onto the MS plate in quadruplicate.

### 2.6. Conditions and Duration of Saliva Storing

In order to establish the appropriate storage conditions for the saliva sample and the duration of time without major alteration to salivary MS profiles, saliva collected with Salivette from four individuals was stored at 4 °C or RT, during 24 h. Twenty microliters of saliva were then prepared as described above (Protocol B) and submitted to MS analysis every hour from 2 to 9 h and 24 h after saliva collection.

### 2.7. Kinetic Collection of Saliva

In order to assess the stability of the intra-individual saliva MS profiles in the short and medium term, kinetic collections were performed with Salivettes. Saliva from four healthy volunteers was collected either at four time points of the same day (8:00 am; 10:00 am; 2:00 pm; 4:00 pm), or during four consecutive days in the morning (between 8:00 am and 10:00 am), or at three times two weeks apart (at day 0 (D0), D15, D30). Salivettes, stored on ice, were either immediately treated for MS analysis or stored at −20 °C for future MS analysis.

### 2.8. Sample Loading for MALDI-TOF MS Analysis

One microliter of each sample was spotted in quadruplicate on MALDI target plate (Bruker Daltonics). After air-drying at RT, 1 µL of matrix solution composed of saturated α-cyano-4-hydroxycinnamic acid (Sigma, Lyon, France), 50% (*v*/*v*) acetonitrile, 2.5% (*v*/*v*) trifluoroacetic acid prepared with HPLC-grade water, was added. Matrix quality (i.e., absence of MS peaks due to matrix buffer impurities) and MALDI-TOF apparatus performance were checked, respectively, by loading only matrix solution and saliva from the same individual prepared with the protocol B, in duplicate onto each MS plate. After drying for a few minutes at RT, the MS target plate was introduced in the mass spectrometer.

### 2.9. MALDI-TOF MS Parameters

Protein mass profiles were obtained using a Microflex LT MALDI-TOF Mass Spectrometer (Bruker Daltonics), with linear positive-ion mode detection at a laser frequency of 50 Hz within the 2–20 kDa mass range. The setup parameters of the MALDI-TOF MS apparatus were identical to those used previously [23]. Briefly, the acceleration voltage was 20 kV, and the extraction delay time was 200 ns. Each spectrum corresponds to ions obtained from 240 laser shots performed in six regions of the same spot and automatically acquired using the AutoXecute of the Flex Control v.3.0 software (Bruker Daltonics).

### 2.10. MS Spectra Analysis

The MS spectra profiles were firstly controlled visually with flexAnalysis v3.3 software (Bruker Daltonics). The MS spectra were then exported to ClinProTools v2.2 and MALDI-Biotyper v3.0. (Bruker Daltonics) for data processing (smoothing, baseline subtraction, peak picking). The reproducibility and specificity of the MS spectra were mainly objectified using the composite correlation index (CCI) tool as previously described [24]. The CCI assessed the variation of the spectra within and between each group of samples as a function of the parameters tested. The CCI matrix was calculated using MALDI-Biotyper v3.0 software with the following default settings: mass range 3000–12,000 Da, resolution 4, 8 intervals and autocorrelation off. A CCI match value of 1 represents a perfect correlation, whereas a CCI match value of 0 represents an absence of correlation. To visualize the numerical values of CCI, a heat-map was automatically generated. The CCI value threshold for establishing MS spectra reproducibility was determined based on previous studies [25] and the lowest CCI values obtained among the technical replicates (quadruplicate) loaded onto the MS plate. Under these conditions, a CCI match value above 0.8, corresponding to a high correlation, confirmed the stability of MS spectra compared.

To visualize changes in MS profiles or variations in MS peak intensity, the gel view tool of ClinProTools v2.2 software was used. 

### 2.11. Statistical Analysis

Statistical analyses were conducted using the GraphPad Prism software 7.0.0 (GraphPad Software, San Diego, CA, USA). Comparison of peak intensity at different times was carried out using a one-way ANOVA, followed by Fisher’s least significant difference (LSD) method to compare each time. As Fisher’s LSD does not account for multiple testing, differences were considered significant at the threshold *p* < 0.01.

## 3. Results

### 3.1. Saliva Sample Preparation

The protocol for pretreatment of saliva sample, prior to loading onto MS plate, was the first parameter tested. For this fine tuning, saliva collected using plastic tubes from four healthy individuals (two males and two females, mean age 45.3 ± 11.9 (±standard deviation, SD)) was used. The volume of saliva collected varied from 1 to 3 mL. The saliva from each individual was prepared with the four distinct protocols (i.e., A, B, C, and D). Two protocols, with a minimum of sample handling, consisted either of direct loading of 1 µL of saliva onto the MALDI plate (Protocol A), or loading after the addition of one volume (i.e., 20 µL) of Mix buffer (Protocol B). The other two protocols (C or D) involved overnight protein precipitation in organic solvent [26] prior to loading onto the MS plate. We observed that the MS profiles were visually reproducible for each individual regardless of the protocol tested (Figure 1a). The high CCI (mean ± SD: 0.87 ± 0.11) obtained between each saliva preparation conditions per individual, showed the reproducibility of the MS spectra (Figure 1b). These results suggest that the MS spectra are little changed regardless of the protocol used for sample pretreatment. As the two organic protocols (i.e., C and D) required a longer sample preparation time, and as homogenization of the precipitated proteins was time-consuming and sometime incomplete, these protocols were not retained. Among the protocols A and B, which are simple and fast (handling time < 2 min per sample), the protocol B was selected and corresponds to the conventional method of sample preparation for MALDI-TOF MS profiling analysis [27].

### 3.2. Effect of Serial Dilution of Saliva on MS Profiles

It is well known that saliva is highly viscous. To facilitate the pipetting of saliva, one volume of ultra-pure water was added to the samples collected in plastic tube. As the amount of saliva was estimated visually, the volume of ultra-pure water added could be overestimated. Moreover, the procurement of adequate volume of saliva could be limited in some individuals such as young children or patients with sialadenitis [28], and sample dilution could be required. Thus, we assessed the consequences on MS profiles of dilutions of saliva with ultra-pure water. Serial dilution of saliva from six individuals (two females and four males, mean age ± SD: 40.3 ± 6.1) with ultra-pure water revealed that the MS profiles remained highly reproducible (Figure 2a, Appendix A). A CCI-based analysis confirmed this reproducibility for all individuals up to the 1/64 dilution, and up to 1/256 for four of them (Figure 2b, Appendix A). Moreover, the low decrease in MS profile intensity with the serial dilutions underlined that large dilutions of saliva appeared to marginally alter the respective MS spectra. 

### 3.3. Saliva Collection Modes

Although the direct collection of saliva in plastic tubes generates high quality MS spectra, some individuals averse this collection method, and it does not prevent the formation of saliva droplets or the flow of saliva out of the collection tube. Moreover, despite a clear explanation of the drooling procedure, some individuals spat out mucous secretions instead of saliva. The heterogeneity of samples could likely generate heterogeneity in the MS profiles. We therefore decided to test a more hygienic collection system, the Salivettes^®^. With this system, a cotton roll, kept few minutes in the participant’s mouth, is wetted with saliva. Five volunteers (three males, two females, mean age ± SD: 41.2 ± 13.5) salivated in plastic tubes and used Salivettes devices in parallel (Figure 3a). The comparison of the protein profiles between these two sampling systems revealed visual similar MS spectra for each individual (Figure 3b,c). However, paired comparisons of CCI values per collection mode for each individual revealed mitigated results. High reproducibility of MS spectra between the two systems was obtained for two individuals (CCI values > 0.85, #1 and #2, Figure 3d). Moderate and low CCI values were obtained for 1 (#4) and two (#3 and #5) individuals, respectively. These results suggested that despite the visual reproducibility of the MS profile replicates between the modes of saliva collection per individual, changes in MS profiles could occur preventing their substitution for saliva collection. For the following experiments, Salivettes, standardizing saliva collection, were used as the saliva collection system.

### 3.4. Effect of Conditions and Duration of Saliva Storing on MS Profiles 

Saliva contains numerous enzymatic proteins involved in the food digestion, which could also induce proteolysis of saliva proteins [29]. Thus, saliva storage conditions and duration of storage are factors which may alter the integrity of the proteins and thereby the respective MS profiles. In order to determine the optimal parameters, saliva from four individuals (three males and one female, mean age ± SD: 38.0 ± 8.0) was collected with the Salivette system and stored either at 4 °C or RT (about 20 °C) and finally loaded onto a MS plate to compare the stability of MS profiles as a function of storage mode and duration. After storage of the saliva samples for 24 h, even at 4 °C, a dramatic change in the MS profiles was observed, confirmed by CCI-based analysis (CCI values < 0.8, Appendix A). The acquisition of the MS spectra can therefore not be performed the consecutive day of sample collection and stored in these conditions. In order to specify the retention periods of saliva samples over one day, sequential analyzes of the MS profiles carried out every hour were compared. On the gel view (Figure 4a), reproducibility of MS spectra was observed, notably for samples stored at 4 °C for more than 8 h. Nevertheless, it seems that a decrease in the intensity of the MS profiles occurred with the increase of the storage time of samples. To assess this phenomenon, the intensities of two MS peaks shared among individuals, P1 and P2 at about 4372 and 6952 *m*/*z*, respectively, were compared. Although the saliva MS profiles from one individual (#4) appeared to be little altered by the duration and storage mode, the overlays of the P1 and P2 MS peaks revealed a decrease in intensity for saliva from three other individuals (#1 to #3) (Figure 4b). The proportion of peak intensity decrease reached more than 89% for P1 (#1 and #3) and 90% for P2 (#1) after 9 h of storage at RT (Figure 4c). Pairwise comparison highlighted significant decreases in peak intensities for P1 and P2 (1-way ANOVA, followed by Fisher’s LSD, *p* < 0.01) starting at about 2–4 h after sample collection at RT compared to 4 °C. Interestingly, the intensity of the P1 and P2 peaks of saliva stored at RT during 4 h were lower than of those of paired samples preserved at 4 °C during 9 h (Figure 4c). These important decreases in peak intensity at RT led to the loss of several MS peaks (Figure 4a). Overall, these results revealed that saliva samples stored at RT could compromise the detection of MS peaks. In this sense, storing of Salivette at 4 °C until sample treatment during the working day appeared as the most efficient conditions prior to saliva analysis by MS profiling.

### 3.5. Intra-Individual Kinetic Evolution of Saliva MS Profiles

Now that the storage and processing conditions of the samples guarantee the reproducibility of the profiles, the stability of the saliva must be verified to ensure it is a reliable indicator. The stability of the salivary MS profiles from four healthy individuals collected on four consecutive days with Salivettes were compared (Figure 5a). The salivary MS profiles remained relatively stable and reproducible over the days for each individual, which was confirmed by high CCI values (ranging from 0.85 ± 0.13 (mean ± SD) to 0.97 ± 0.02) (Figure 5b). To extend the analysis to longer timescales, saliva samples were collected from six individuals at three time points with an interval of two weeks, and the resulting salivary MS profiles were compared (Appendix A). As observed for consecutive collection days, saliva MS profiles remained visually stable over the month (Appendix A), which was supported by the high CCI values obtained for each individual (ranging from 0.81 ± 0.31 (mean ± SD) to 0.97 ± 0.04, Appendix A).

### 3.6. Low Variation of Saliva MS Profiles during the Day (Circadian Variability)

The saliva samples used in the above experiments were all collected in the morning between 8:00 and 10:00. However, depending on the time of day or satiety, variations in the saliva composition may occurred [29]. To assess whether saliva variations over the course of one day could impact on MS profiles, saliva was collected with Salivettes from four healthy volunteers (four men and one woman, mean age ± SD: 36.2 ± 8.1) at four time points (8:00, 11:00, 14:00, and 16:00) of the same day. Except for individual #5, the gel view (Figure 6a) as well as the CCI values (Figure 6b) revealed good reproducibility of the saliva MS profiles per individual between the different time points. For the individual #5, changes were observed in the MS profile of saliva collected at 08:00 compared to the other three time points on the same day. Interestingly, the MS spectra from other three time points (i.e., 11:00, 14:00, and 16:00), were highly reproducible with a CCI value of 0.97 ± 0.03. These results suggested that the individual #5 likely performed an improper mouth washing prior to saliva collection for the primary sampling of the day, which explains this unique time point variation. Collectively, these results supports that the time of collection during the day did not seem to alter the reproducibility of the whole saliva MS profiles if proper mouth washing was performed.

### 3.7. Heterogeneity of Saliva MS Profiles among Healthy Individuals 

Now that all the analysis and collection parameters allow reliable profiles to be obtained, it is possible to extend the cohort of individuals and compare their profiles. The saliva MS profile of 12 healthcare workers was measured. The comparison of the profiles indicates that although several MS peaks were shared, MS profiles appeared to be rather singulars (Figure 7a). The very low CCI values obtained by pairwise comparisons of some individuals, ranging from 0.02 ± 0.00 (mean ± SD) to 0.88 ± 0.03, confirmed the high inter-individual heterogeneity of saliva MS profiles (Figure 7b).

## 4. Discussion

The repeated demonstration of detection of SARS-CoV-2 in the saliva from individuals diagnosed with COVID-19 [30] and its transmission by talking, sneezing, and coughing due to saliva droplets production [31] indicated that saliva is suitable for virus detection. Prior to assess the potential of the MALDI-TOF MS profiling approach for diagnosing SARS-CoV-2 in saliva, it is required to establish a standardized operational protocol. Indeed, for a same individual, protein repertories are clearly distinct according to sample sources [32,33], and resulting MS profiles are directly linked to the sample preparation modes and storage conditions [34]. In the present work, the consequences of saliva sample management, from collection to MS analysis, on MS protein profiles were analyzed using saliva from healthy individuals. As the ultimate goal will be to detect changes in MS profile as a function of individual SARS-CoV-2 infection status, it is necessary to limit changes in MS profiles due to external factors or inappropriate management. Throughout the tests performed here, the primary criterion monitored was the intra-sample reproducibility of MS spectra. Concordant results were obtained for the large majority of the tests performed among biological replicates assessed in the same conditions. Nevertheless, as the number of individuals successfully enrolled per experiments was limited, the use of a larger cohort could confirm these results.

For preparing the samples for MALDI-TOF analysis, among the four protocols tested, the fastest protocol requiring the least amount of handling was selected. It consisted of mixing (*v*/*v*) one volume of saliva sample with the conventional Mix buffer used for MALDI-TOF [23]. Its simplicity and the use few and inexpensive reagents are key factors for mass-testing [35]. For saliva collection, the Salivette device was privileged as it presents numerous advantages. It is a hygienic system that avoids the formation of saliva droplets and allows self-collection, preventing exposure of healthcare workers [36]. Conversely to saliva collection by spitting into a plastic tube, where mucous secretions (i.e., sputum) could be mixed with saliva, the Salivette allows standardized collection [37]. Limitations to the use of the Salivette are the price of the system (about $1) compared to direct drooling in a plastic tube (few cents) and the risk of a shortage of these devices, as it has been reported for NPSs [38,39]. Interestingly, although saliva collected by Salivette or plastic tubes resulted in visually similar protein profiles, an accurate comparison of CCI values indicated that reproducibility was variable among individuals. Previous studies already reported that the performance of protein biomarker detection in saliva was dependent on the mode used for saliva collection [40]. 

The high throughput capability of MALDI-TOF MS profiling makes this tool well suited for mass-testing [41]. While a small volume of saliva (20 µL) is required for MS analysis, validation of results by molecular biology requires larger sample volumes (150–300 µL) [42]. The Salivette system permits the collection of up to 2 mL of saliva [43]. However, the amount of saliva retrieved varies between individuals. It has recently been reported that up to 10% of individuals failed to soak the cotton roll sufficiently for future molecular analyses and the respective saliva samples were then excluded from the study [37]. To limit the exclusion of saliva samples, it has been proposed that ultra-pure water be added to the cotton roll, which does not compromise RNA detection, including SARS-CoV-2 and the human RNase P, an internal cellular control [22]. Here, we demonstrate that dilution of the saliva sample with ultra-pure water to 1/64 could be performed without altering the resulting MS profiles. Thus, the collection of a small volume of saliva with the Salivette system appears compatible for molecular analysis and MS profiling.

Whole saliva contains numerous enzymatic proteins involved in the food digestion, which could also induce lysis of saliva proteins [44,45]. To inhibit enzymatic activity, anti-protease cocktails are generally used [46]. However, the efficiency of these digestive protein inhibitors in whole saliva appeared insufficient to prevent salivary protein degradation [45]. An evaluation of the storage conditions of saliva samples for various durations revealed a rapid degradation of MS profiles and a decrease in intensity of MS peaks in about 4 h after collection for samples stored at RT. Conversely, samples stored on ice significantly limited protein degradation, preserving MS profiles until 9 h after collection. Therefore, saliva samples should be stored at 4 °C and processed on the same day of collection.

Now that optimal conditions have been established for maintaining the integrity of protein MS profiles during saliva collection and handling, the reproducibility of saliva MS profiles in healthy individuals over time is a key factor to assess, prior to future investigation of biomarkers associated to SARS-CoV-2 infection. Although, salivary protein abundance variations could occurred with circadian rhythm [47], we obtained reproducible salivary MS profiles for each individual collected at successive times of the same day. Salivary proteins possessing a circadian expression were generally discovered on pre-fractionated samples or methods (e.g., immune-precipitation) targeting molecule(s) of interest [47,48]. Their abundances and variations were insufficient for direct detection in whole saliva. The stability of MS profiles over the day per participant emphasized a moderate effect of circadian rhythm on the protein repertoire of whole saliva. Thus, saliva collection could be performed throughout the day without altering the respective MS spectra.

The detection of biomarkers by MS profiling in saliva corresponding to a change in physiological state (transition from healthy to infected status) or symptomatology (recovery or deterioration of physiological state) in COVID-19 patients implies the identification of specific protein signatures. The investigation of MS profile evolution from kinetic collection of saliva from healthy individuals revealed a strong correlation of MS spectra per individual, either for successive daily collections of saliva or for saliva collected two weeks apart. We observed that, for the majority of individuals, the profiles remain globally stable between these time points. Stability of salivary protein profiles was previously reported for the same individual over a five-month period using roll cotton for saliva sampling [49]. It can be therefore hypothesized that in the event of a modification in the physiological state of individuals, a change in protein profile could emerge and would then be detected by MS as described previously [8].

Interestingly, despite the sharing of some MS peaks between healthy individuals, strong heterogeneity was observed. The inter-individual variations confirmed that strategies commonly used in microbiology to classify samples based on homology of MS spectra (matching) with a reference spectra library could not be applied [50]. The identification of specific protein signature associated with COVID-19 infection status will require advanced bio-statistical analyses and the development of machine learning approaches [51].

## 5. Conclusions

On the basis of the results of the present work, we recommend the following guidelines for MS profiling analyses of saliva samples. The Salivette device should be used to homogenize the saliva collection. The collected saliva must be stored in ice before being mix with the conventional mix buffer and loaded onto a MS steel plate on the day of collection. These recommendations allow for intra-individual stability of saliva MS profiles, which is the key factor for future investigation of biomarkers associated to SARS-CoV-2 infection. However, the singularity of MS profiles among individuals suggests that the recourse to advance computational analyses appears ineluctable. In any case, MALDI-TOF MS profiling of human saliva remains a tremendous tool, without PCR, for screening SARS-CoV-2.

## Figures and Tables

**Figure 1 diagnostics-11-01304-f001:**
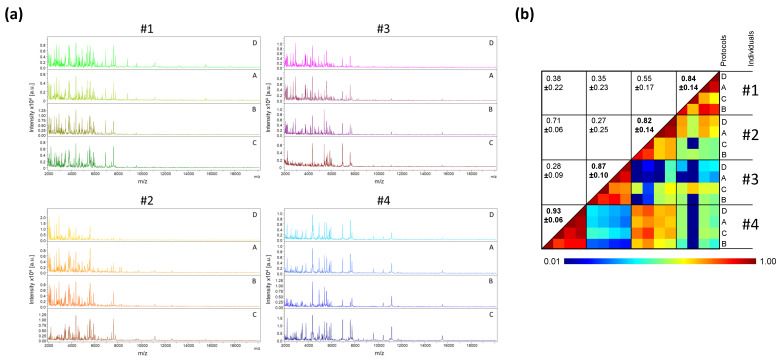
Comparison of MALDI-TOF MS profiles from saliva samples pre-treated with distinct protocols. (**a**) Representative MS spectra from saliva, collected in plastic tube, of four healthcare workers (#1 to #4) per condition of preparation (protocols A to D). a.u., arbitrary units; *m*/*z*, mass-to-charge ratio. (**b**) Assessment of saliva spectra reproducibility for sample pre-treated with distinct protocols using composite correlation index (CCI). The levels of MS spectra reproducibility are indicated in red and blue, revealing relatedness and incongruence between spectra, respectively. CCI are expressed as the mean ± standard deviation. CCI-values per individuals among the different protocols were indicated in bold. A: Saliva diluted with 1 vol. of water; B: A + 1 vol. of Mix buffer; C: 100 µL of A + 400 µL absolute ethanol at −20 °C overnight (ON), centrifugation and pellet re-suspended with 100 µL of Mix Buffer; D: 100 µL of A + 400 µL ACN at −20 °C ON, centrifugation and pellet re-suspended with 100 µL of Mix Buffer. #1–4: healthcare workers.

**Figure 2 diagnostics-11-01304-f002:**
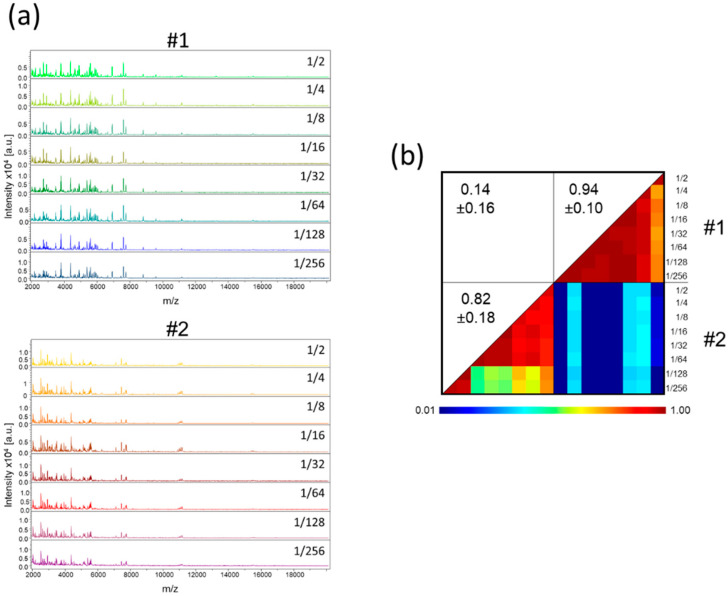
MS profiles of serial dilution of saliva from healthcare workers. (**a**) Representative MS spectra of serial dilution of saliva, collected in plastic tube, prepared according to protocol B from two individuals. The dilution rate is indicated at the right part of each profile. (**b**) Composite correlation index (CCI) matrix representing the levels of MS spectra reproducibility between saliva samples. CCI are expressed as the mean ± standard deviation. #1 and #2: healthcare workers.

**Figure 3 diagnostics-11-01304-f003:**
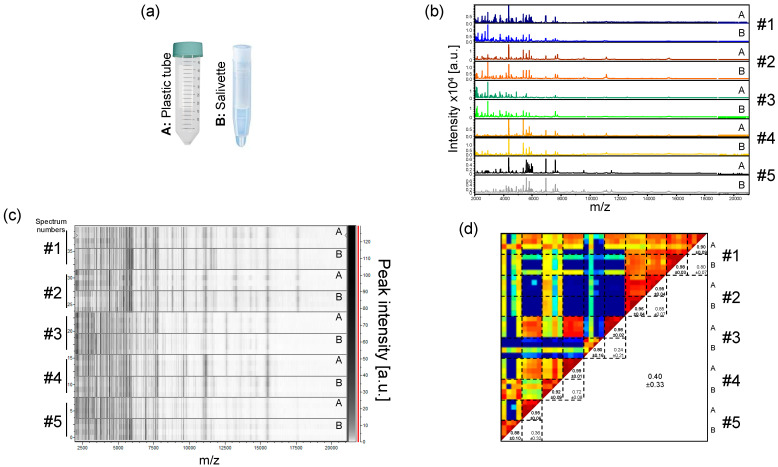
Comparison of saliva MS profiles from healthcare workers collected with two distinct systems. (**a**) Representation of the two saliva collection systems used. (A) Plastic tube; (B) Salivette^®^. (**b**) Representative saliva MS spectra collected either directly in plastic tube (A) or in a Salivette system (B) from five healthcare workers. (**c**) Gel view of the saliva MS profiles from the same five healthcare workers. The four replicates loaded on the MS plate for each individual per saliva collection system are presented. The gray scale in the right part of the panel correspond to the MS peak intensity level in arbitrary units (a.u.). (**d**) Composite correlation index (CCI) matrix representing the levels of MS spectra reproducibility between saliva samples. CCI are expressed as the mean ± standard deviation. #1 to #5: healthcare workers.

**Figure 4 diagnostics-11-01304-f004:**
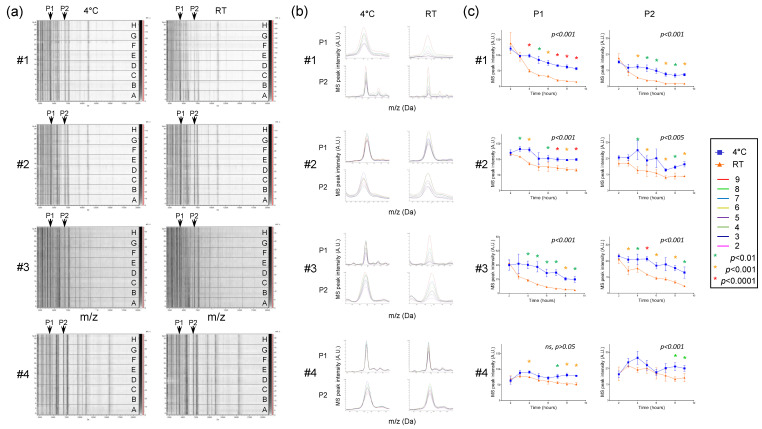
Assessment of storing mode and duration on saliva MS profiles. (**a**) Gel view of saliva MS spectra collected with Salivette and stored either at 4 °C or room temperature (RT) during 2, 3, 4, 5, 6, 7, 8, or 9 h (A to H, respectively). The four replicates loaded on the MS plate for each individual per condition are presented. Two selected MS peaks, P1 (*m*/*z*: 4372 Da) and P2 (*m*/*z*: 6952 Da) are indicated by arrows. #1 to #4: healthcare workers. (**b**) Overlay profile view of selected MS peaks (P1 and P2) from kinetic collection according to storing mode. Line color code of storing time is indicated in the left part. (**c**) Graphical representation of the intensity of selected MS peaks (P1 and P2) according to duration and storing mode. One-way ANOVA, followed by Fisher’s LSD tests were done. *p*-values of one-way ANOVA were indicated on each graphic. Significant differences in peak intensities were indicated by colored asterisks (Fisher’s LSD). Standard deviations of intensities are represented by vertical lines. A.U.: arbitrary units; *m*/*z*: masse to charge ratio.

**Figure 5 diagnostics-11-01304-f005:**
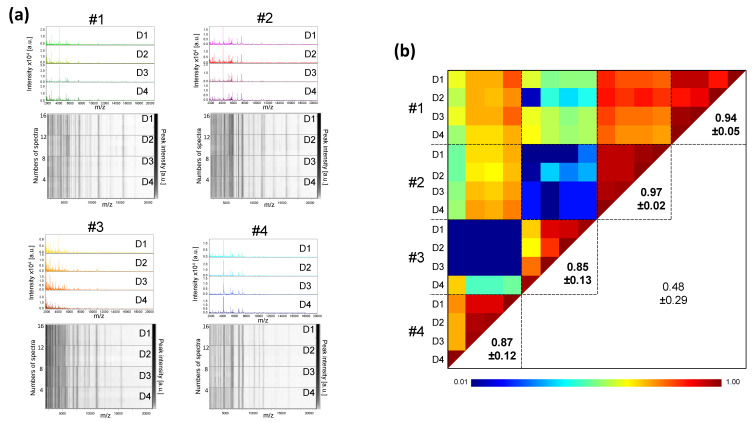
Inter-day reproducibility of saliva MS profiles from healthy individuals. (**a**) Representative saliva MS spectra and gel view of saliva MS spectra from four distinct healthy individuals (#1 to #4) collected with Salivette devices during four successive days (Day, D1, D2, D3, and D4). Per collection day, one MS spectra and the four replicates loaded on the MS plate for each individual are presented on the graphic and gel view, respectively. The gray scale in the right part of the panel correspond to the MS peak intensity level in arbitrary units (a.u.). (**b**) Composite correlation index (CCI) matrix representing the levels of MS spectra reproducibility between saliva samples. Bold numbers correspond to CCI values obtained for each individual in successive days. CCI are expressed as the mean ± standard deviation. #1 to #4: healthcare workers.

**Figure 6 diagnostics-11-01304-f006:**
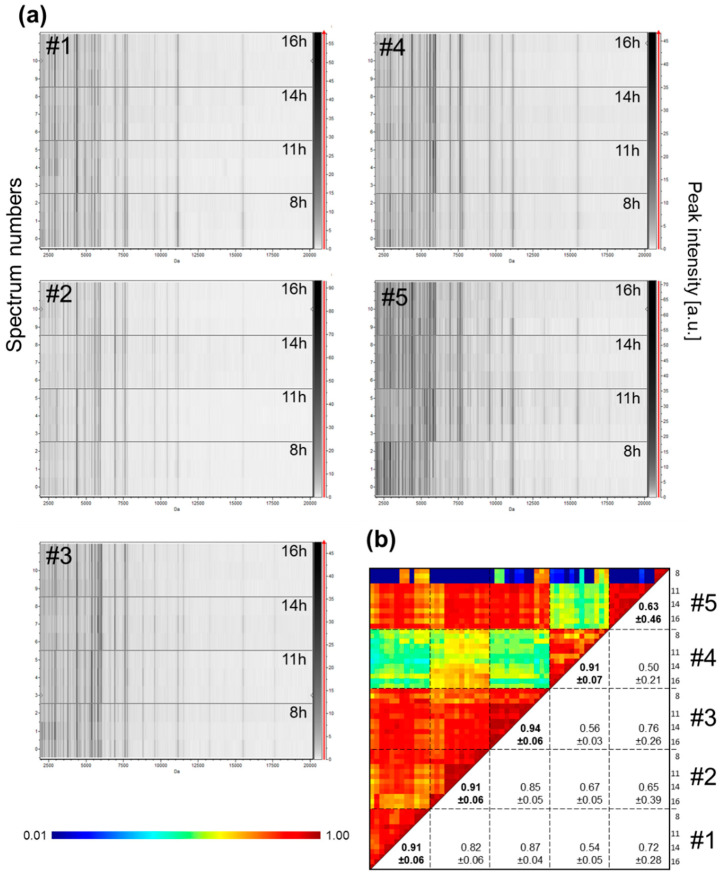
Evolution of MS profiles from kinetic collection of saliva in the same day from healthcare workers. (**a**) Gel view of saliva MS spectra from five distinct individuals. The four collection time points per individual during the same day are indicated in the right part of the gel view. The four replicates loaded on the MS plate for each individual per time point are presented. (**b**) Composite correlation index (CCI) matrix representing the levels of MS spectra reproducibility between collection time points per individual. The values correspond to the mean coefficient correlation and respective standard deviations obtained for paired condition comparisons. CCI are expressed as the mean ± standard deviation. #1 to #5: healthcare workers.

**Figure 7 diagnostics-11-01304-f007:**
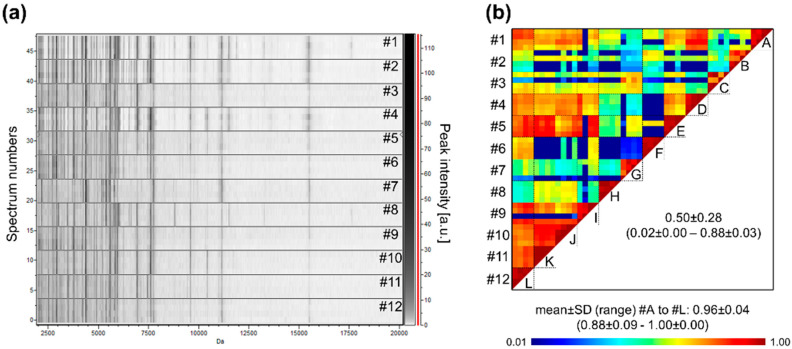
Saliva MS profiles from 12 healthcare workers. (**a**) Gel view of saliva MS spectra from 12 distinct individuals. The four replicates loaded on the MS plate for each individual per are presented. (**b**) Composite correlation index (CCI) matrix representing the levels of MS spectra reproducibility between individuals. The values correspond to the mean coefficient correlation and respective standard deviations obtained for paired condition comparisons. CCI are expressed as the mean ± standard deviation. #1 to #12: healthcare workers. A to L: assessment of individual MS spectra reproducibility.

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
