# Peer review of "Optimization and Standardization of Human Saliva Collection for MALDI-TOF MS"

_diagnostics, 2021, doi:10.3390/diagnostics11081304_

Round 1

Reviewer 1 Report

It's not entirely clear how the subject matter of the article is consistent with SARS-CoV-2? We are talking about miss-spectrometric profiles and their reproducibility in different ways of collecting saliva. However, this information is not new and has long been described in the literature (for example, https://doi.org/10.1016/j.trac.2019.115781). It seems to me that it is necessary to place the emphasis differently in the article on more valuable information about the variability of the composition and ways to reduce the scatter of data, and not SARS-CoV-2. I did not understand why the protocols are named starting with D. The conclusions made on the basis of 4 volunteers are not correct, in my opinion. 

Author Response

Responses to reviewer 1 comments:

Reviewer 1: It's not entirely clear how the subject matter of the article is consistent with SARS-CoV-2? We are talking about miss-spectrometric profiles and their reproducibility in different ways of collecting saliva. However, this information is not new and has long been described in the literature (for example, https://doi.org/10.1016/j.trac.2019.115781). It seems to me that it is necessary to place the emphasis differently in the article on more valuable information about the variability of the composition and ways to reduce the scatter of data, and not SARS-CoV-2. I did not understand why the protocols are named starting with D. The conclusions made on the basis of 4 volunteers are not correct, in my opinion.

Response: We agree with the reviewer that, as none of the individuals included in the present study were infected by SARS-CoV-2, the relationship between this work and infection with this coronavirus was initially not evident. This study was conducted at the beginning of the SARS-CoV-2 pandemic and the aim was to develop an original, PCR-free approach for SARS-CoV-2 diagnosis (according to the CoviDiagMS Project, Grant n°2020-COVID19-15). We decided thus to explore the potential of MALDI-TOF MS profiling to classify SARS-CoV-2 positive from negative individuals. Moreover, to facilitate sampling for patients and health care workers, we decided to use saliva as specimens. We recently demonstrated that saliva collection with Salivette® improved SARS-CoV-2 diagnosis from symptomatic and asymptomatic patients compared to NPS (Melo Costa et al, J Oral Microbiol. 2021). Nevertheless, to avoid any confusion, the name of SARS-CoV-2 was not included in the manuscript title.
The reviewer is correct that previous works has already described the advantages and limitations of different methods of saliva collection, notably for proteomic analyses. But to our knowledge, no work has established the optimal conditions for sampling, storing, and managing saliva for future analysis by MALDI-TOF MS profiling for human diagnosis.
The publication from Bellagambi et al. (Trends in Analytical Chemistry, 2020) mentioned by the reviewer is a very interesting review describing the production and composition of saliva, the different ways of saliva collection and finally the detection methods for identifying salivary biomarkers of clinical interest. Unfortunately, we missed this article probably because it was not referenced in PubMeb. This article is now referenced in our manuscript (discussion section).
Nevertheless, this review presented studies in which the proteomic analysis was focused on the detection of specific biomarkers, i.e. the protein identity or peak list (mass/charge ratio) is already known. In contrast, MALDI-TOF MS profiling is based on the acquisition of a complete MS profile of saliva samples, in order to subsequently search for a protein signature associated to a specific character (such as SARS-CoV-2 infection). The originality of the present work lies in the description of the successive steps carried out to determine the best conditions to collect and prepare saliva samples for the MS analyses used in MALDI-TOF MS profiling. The search for a protein signature associated with SARS-CoV-2 infection is not addressed here. It is the subject of a second manuscript, in progress, linked to this one, underlining the potential of MALDI-TOF MS profiling associated to machine learning models to classify SARS-CoV-2 patients based on the analysis of saliva samples.
We recognized that the number of volunteers included in each tests was modest, and this point was also underlined by the reviewer two. Details about this point were already done in response to comment 4 from reviewer 2. According to the expectations of both reviewers, the results obtained from the others individuals tested were either directly included to the figures already presented or added in supplementary data. For figure 1, figure 4 and figure 5, the results from 4 individuals were available and are then presented. It is
possible to perform supplementary experiments by including more individuals, but in this condition, we need more time to plan individual convocation and to realize new experiments. A sentence was added in the discussion section emphasizing that the number of individual enrolled was limited.
Concerning the names of the protocols, the first one was called “D” because it corresponds to a “Direct” loading of the saliva onto the MS plate without any treatment. Effectively, it could be confused that the first one presented in the Materials and Methods section was named “D” and not “A”. This point was now corrected throughout the manuscript.

Reviewer 2 Report

SARS-CoV-2 outbreak led to unprecedented scientific innovative research to preclude the virus dissemination and limit its impact on life expectancy. Waiting for the collective immunity by vaccination, mass-testing and isolation of positive cases remain essential. The development of a diagnosis method requiring a simple and non-invasive sampling with a quick and low-cost approach is on demand.

The authors hypothesized that the combination of saliva specimens with MALDI-TOF MS profiling analyses could be the winning duo.

Before characterizing MS saliva signatures associated with SARS-CoV-2 infection, optimization and standardization of sample collection, preparation and storage up to MS analyses appeared compulsory. In this view, successive experiments were performed on saliva from healthy healthcare workers.

Specimen sampling with a roll cotton of Salivette® devices appeared the most appropriate collection mode. Saliva protein precipitation with organic buffers did not improved MS spectra profiles compared to a direct loading of samples mixed with acetonitrile/formic acid buffer onto MS plate. The assessment of sample storage conditions and duration revealed that saliva should be stored on ice until MS analysis, which should occur on the day of sampling. Kinetic collection of saliva highlighted reproducibility of saliva MS profiles over four successive days and also at two-week intervals.

The authors concluded that the intra-individual stability of saliva MS profiles should be a key factor in the future investigation for biomarkers associated with SARS-CoV-2 infection. However, the singularity of MS profiles between individuals will require the development of sophisticated bio-statistical analyses such as machine learning approaches. MALDI-TOF MS profiling of saliva could be a promising PCR-free tool for SARS-CoV-2 screening.

Comments:

1. Overall, it is an important and thoroughly and systematically well conducted work, however there are some (minor) shortcomings, that should be corrected in the manuscript, or stated in the limitations. The major weakness of the paper is that, it promises new technologies for detecting SARS-CoV-2 but they present data only on healthy individuals.

2, The Introduction is written in a satisfactory manner. However little is said about the MALDI-TOF profiling technique. Advantages and disadvantages are not presented. The hypothesis is vaguely described, This part should be more focused.

3. The Materials and Methods are well written. However, criteria for comparison of various methods (saliva collection, saliva processing) are not stated, although the results contain some of these criteria.

4. Although the number of participants is stated, it is unclear how these 12 participants were distributed across the groups (Drooling in Plastic Tubes vs Cotton roll system; the fourSaliva Sample Preparation protocols, etc.).

5. All investigated individuals were healthy so their saliva does not have the same conditions as the ones you have to dilute. Hence no evidence is provided that dilution has no effect on conditioned saliva samples (For example People having dense saliva, or fever, or sialadenitis this is not true).

6. The n=2 is very low and unacceptable for quantitative studies

7. In the Discussion the limitations of the study is missing.

8. Statistical analysis was conducted in only a few instances for comparing MS protein profile. Statistics should be applied in all experiments,

9. The low number of sample sizes in each investigated group should prevent any quantitative conclusions. Sample size need to be increased to n=5 in each experiment.

10. Among the abbreviations MALDI-TOF MS is missing from the abbreviations

Author Response

Responses to reviewer 2 comments:

1. Overall, it is an important and thoroughly and systematically well conducted work, however there are some (minor) shortcomings, that should be corrected in the manuscript, or stated in the limitations. The major weakness of the paper is that, it promises new technologies for detecting SARS-CoV-2 but they present data only on healthy individuals.

Response: This comment is similar to that made by the reviewer 1. The reviewer is correct, in this study, uniquely healthy individuals were included. This work is linked to a second one which will be submitted soon aiming to demonstrate the potential of MALDI-TOF MS profiling for SARS-CoV-2 diagnosis using saliva. The success of this classification was attributed in part to the establishment of the better conditions for the collection, management and storing of saliva samples prior to searching for a protein signature associated to SARS-CoV-2 infection. The optimized procedures will be applied to research salivary protein signatures following alteration of physiological conditions, this point was already underlined in the discussion section. Moreover, as this work correspond to preliminary tests from the CoviDiagMS Project (Grant n°2020-COVID19-15), it is necessary to detail its relationship to SARS-CoV-2 infections to comply with the conditions of the grant.

2. The Introduction is written in a satisfactory manner. However little is said about the MALDI-TOF profiling technique. Advantages and disadvantages are not presented. The hypothesis is vaguely described, This part should be more focused.

Response: According to reviewer, some sentences were added in the introduction section to detail MALDI-TOF MS profiling principle and to present some advantages of this tool. The limitations of the approach were indicated in the discussion section. Concerning the hypothesis of the use of MALDI-TOF MS profiling for future SARS-CoV-2 diagnosis, it was done, as indicated, based notably on the previous works presented above, demonstrating the performances of this tool “to classify individuals according to their infectious status” using saliva as specimens, but also for detection of infected viral cell culture. The detection of SARS-CoV-2 in saliva from infected patients by molecular and proteomic approaches underlined the interest to select this biological fluid. Finally, to overcome the molecular limitations of SARS-CoV-2 diagnosis, this PCR-free approach appears as a relevant alternative. All these points were already included in the introduction section, so we does not know which information is still missing about the hypothesis proposed.

3. The Materials and Methods are well written. However, criteria for comparison of various methods (saliva collection, saliva processing) are not stated, although the results contain some of these criteria.

Response: To avoid redundancy in the Materials and Methods section, the criteria used for comparison of MS spectra in the different experiments were indicated in the paragraph “2.10. MS spectra Analysis” and “2.11. Statistical Analysis”. The main criteria tested were the reproducibility and stability of the MS spectra. To assess the level of MS spectra reproducibility, the Composite Correlation Index (CCI) was systematically computed. The CCI matrix, a statistical autocorrelation method, was constructed to measure the relationships between all spectra. A CCI match value of 1 represents perfect correlation, whereas a CCI match value of 0 represents an absence of correlation. The CCI value threshold for establishing MS spectra reproducibility was determined based on previous studies (Murugaiyan J. et al., Frontiers in Microbiology, 2018) and the lower CCI values obtained among the technical replicates (quadruplicate) loaded onto MS plate. Here, the CCI values among the quadruplicate per individual and condition varied from (range+/-SD) 0.80±0.16 to 1.00±0.00. In such condition, a CCI match value upper than 0.8 corresponding to a high correlation, confirmed the stability of MS spectra compared. This point was added in the materials and methods section.

4. Although the number of participants is stated, it is unclear how these 12 participants were distributed across the groups (Drooling in Plastic Tubes vs Cotton roll system; the four Saliva Sample Preparation protocols, etc.).

Response: To carry out these different tests, a total of 12 individuals were enrolled. Unfortunately, they were rarely all present when called for sampling. Then, the distribution for each experiment was more associated with subjects present than with any particular selection. We could consider the inclusion as non-supervised. For each the experiments, 4 to 12 individuals were then included. The results of some individuals were not included initially in the manuscript because these results were concordant with those presented in the manuscript and the addition of these data induced overloading of the illustrations (MS profiles, gel views, graphics) altering their legibility. That’s why, uniquely representative results were presented. Then, the results obtained from the others individuals tested are now added either directly in the respective figure or in additional file to avoid alteration of the readability of the figures. Unfortunately, for the experiment from the figure 1, 4 and 5, solely 4 individuals were present. As indicated above in the answer to reviewer 1, it is possible to add new individuals if the time offered to bring in new experiments was extended. All these modifications were done in the figures, figure legend and result section directly in the manuscript.

5. All investigated individuals were healthy so their saliva does not have the same conditions as the ones you have to dilute. Hence no evidence is provided that dilution has no effect on conditioned saliva samples (For example People having dense saliva, or fever, or sialadenitis this is not true).

Response: We apologize but we are not sure to correctly understand this comment. Saliva dilution was tested because, as indicated in the manuscript (Result section “3.2. Effect of serial dilution of saliva on MS profiles”), the high viscosity of saliva disturbs correct pipetting, and the addition of ultra-pure water could facilitate pipetting. To avoid this viscosity problem and to homogenize the saliva samples, Salivettes were finally selected as underlined in the manuscript (Result section “3.3. Saliva Collection Modes”). Moreover, it was reported (Aita et al, Clin. Chim. Acta Int. J. Clin. Chem. 2020) that the volume of saliva collected with Salivette was insufficient for 10% of the SARS-CoV-2 patients, who were therefore excluded from the study. In a previous study, we confirmed that the saliva volume retrieved with Salivettes was insufficient for 11% (34/303) of the saliva collected (Melo Costa et al, J Oral Microbiol. 2021). However, in this last previous study, solely 5 of 34 were diagnosed as SARS-CoV-2 positives underlining that the miss-salivation could not be attributed to infectious status. Effectively, the proportion of samples with insufficient saliva retrieved were similar in the SARS-CoV-2 positive and negative groups, emphasizing that the infectious status was not linked to the salivation performances. Saliva dilution was performed to assess whether it could alter resulting MS profiles. Here, we observed no change in MS profiles diluted until 1/64 as confirmed by the high values of CCI. The inclusion of the results from four additional individuals confirmed this reproducibility, which are presented in the supplementary file Figure S1 (new figure). We agree with the reviewer that it is possible that saliva composition could be altered following changes in the physiological state of the individual, which is the hypothesis for the use of MS profiling for SARS-CoV-2 diagnosis (research of a specific protein signature). However, we did not understand why some MS peaks will be suppressed following the dilution of these saliva. Moreover, for your information, our preliminary results revealed that we obtained a reproducibility of MS profiles after saliva dilution regardless of SARS-CoV-2 infectious status (unpublished data). Then, no specific change was done concerning this point.

6. The n=2 is very low and unacceptable for quantitative studies

Response: The reviewer is correct. As indicated above (responses to general comment from reviewer 1 and comment 4 from reviewer 2), the number of individuals included in each experiment were generally larger. All the individuals tested were then now added to the respective figures or are available in the supplementary files section to keep the readability of the illustrations.

7. In the Discussion the limitations of the study is missing.

Response: The main limitation of the present work is the absence of SARS-CoV-2 samples to control whether an evident protein signature could emerge following viral infection. Nevertheless, the comparison of MS profiles highlighted an important heterogeneity among healthy individuals and then the need for sophisticated bio-informatics analyses appears compulsory. This point was already mentioned at the end of the discussion section. The others limitations of the present study are the low number of individuals present for some experimental tests as underlined by the reviewers. For the majority of the experiments, more than five individuals were included, nevertheless, for few of them the number of individuals collected were lower. Although, solely 4 individuals were included to assess the reproducibility of the saliva MS profiles per individuals on 4 consecutive days (Figure 5), the saliva MS profiles of 6 individuals collected at two week of interval during one month were presented (Figure S3). On these both analyses, we observed a high reproducibility of the saliva MS profiles corroborating the stability of these profiles for healthy individuals at short and medium term. For the testing of the mode and duration of sample storing, only four individual were tested. However, the concordance of the results obtained between these four individuals and the corroboration with the literature (Ruhl S. Expert Rev Proteomics. 2012; Thomadaki K, et al. J Dent Res. 2011) reporting proteolysis of saliva samples, support our conclusion. Nevertheless, the low number of individuals included in the present study was underlined in the discussion section.

8. Statistical analysis was conducted in only a few instances for comparing MS protein profile. Statistics should be applied in all experiments,

Response: The majority of the experiments were performed to assess whether the different tests induce MS protein changes, according to collection, storing, management or also for kinetic sampling. A relevant method to evaluate MS spectra reproducibility is the used of the Composite Correlation Index (CCI). Details about the criteria to identify highly reproducible MS spectra were now added in the materials and methods section (“2.10. MS spectra Analysis”) and in the response to the comment 3 from the reviewer 1.

9. The low number of sample sizes in each investigated group should prevent any quantitative conclusions. Sample size need to be increased to n=5 in each experiment.

Response: The reviewer is correct. This point was already largely detailed in the responses to reviewer 1 and the responses to comments 4 and 6 from reviewer 2, as indicated above. Moreover, a sentence was added in the discussion section about the limited number of individual enrolled for few experiments.

10. Among the abbreviations MALDI-TOF MS is missing from the abbreviations

Response: The abbreviation of MALDI-TOF MS was added in the manuscript and the abbreviation list according to reviewer comment.

Round 2

Reviewer 1 Report

The authors took into account the comments of the reviewers and significantly revised the manuscript. However, it seems to me inappropriate to focus on SARS-CoV-2 in the annotation and introduction. If I were the authors, I would change the emphasis, the article is of interest to readers without it. 

Author Response

"The authors took into account the comments of the reviewers and significantly revised the manuscript. However, it seems to me inappropriate to focus on SARS-CoV-2 in the annotation and introduction. If I were the authors, I would change the emphasis, the article is of interest to readers without it."

The reviewer is correct and the protocol established here could be applied for others analyses using MALDI-TOF MS besides the COVID-19 diagnosis. However, as we explained in our previous answer, it is necessary to keep the association with SARS-CoV-2 for several reasons. In one hand, a second manuscript, under progress, demonstrated the performance of MALDI-TOF MS approach for SARS-CoV-2 diagnosis, based on saliva specimens applying the standardized conditions presented here. The optimization steps established in the current study, were fundamental to guarantee the accuracy of sample classification and to limit MS spectra variations attributed to methodological procedures. In addition, the standardization of sampling and sample management allowed to access the level of heterogeneity that could be found in saliva samples or other factors that could affect MALDI-TOF MS analyses, before any characterization of the protein signature associated with SARS-CoV-2 infection, as was mentioned in the manuscript. Moreover, as it was underlined in the original version and noticed in the modifications performed in the  first review of this study, It is clearly indicated that this protocol could be applied to others kind of analyses combining saliva collection and MALDI-TOF MS.

In the other hand, as this research was performed to establish the standardize conditions for saliva collection and management for future SARS-CoV-2 diagnosis by MALDI-TOF MS, it appears logical to refer to this coronavirus in this manuscript, notably for the other teams which would like applied this approach for COVID-19 mass-testing. Moreover, this work was supported by a grant obtained for the CoviDiagMS project and our works should then done in relation with this coronavirus.

Reviewer 2 Report

The replies of the authors are satisfactory.

Author Response

We appreciate the considerations from reviewer 2 regarding our changes in the manuscript and acceptance for publication.